



# Multi-spatial scale assessment and multi-dataset fusion of global terrestrial evapotranspiration datasets

Yi Wu , Chiyuan Miao*, Yiying Wang, Qi Zhang, Jiachen Ji, Yuanfang Chai

State Key Laboratory of Earth Surface Processes and Resource Ecology, Faculty of Geographical Science, Beijing Normal University, Beijing 100875, China.

*Correspondence to*: Chiyuan Miao (miaocy@bnu.edu.cn)

**Abstract.** Evapotranspiration (ET) is an important component of the terrestrial water cycle, carbon cycle, and energy balance. Currently, there are four main types of ET datasets: remote sensing–based, machine learning–based, reanalysis–based, and

land–surface–model–based. However, most existing ET fusion datasets rely on a single type of ET dataset, limiting their ability to effectively capture regional ET variations. This limitation hinders accurate quantification of the terrestrial water balance and understanding of climate change impacts. In this study, the accuracy and uncertainty of thirty ET datasets (across all four types) are evaluated at multiple spatial scales, and a fusion dataset BMA(Bayesian model averaging)-ET, is obtained using BMA method and dynamic weighting scheme for different vegetation types and non-common cover years

among ET datasets. ET from FLUXNET as reference, the study recommends remote sensing– and machine learning–based ET datasets, especially Model Tree Ensemble Evapotranspiration (MTE) and Penman-Monteith-Leuning (PML), but the optimal selection depends on season and vegetation type. At the basin scale, land–surface–model–based ET datasets have less relative uncertainty compared to other types of ET. At the global scale, the uncertainty is lower in regions with larger ET, such as the Amazon, Central and Southern Africa, and Southeast Asia. The BMA-ET dataset accurately captures trends and

seasonal variability in ET, showing a global terrestrial increasing trend of 0.21 mm·yr$^{-1}$ over the study period. BMA-ET has higher correlation coefficients and lower root-mean-square errors than most individual ET datasets. Validation using ET from FLUXNET as reference shows that correlation coefficients of more than 70% of the flux sites exceed 0.8. Overall, BMA-ET provides a comprehensive, long-term resource for understanding global ET patterns and trends, addressing the limitation of prior ET fusion efforts. Free access to the dataset can be found at

https://doi.org/10.6084/m9.figshare.28034666.v1 (Wu and Miao, 2024).

**Keywords:** Evapotranspiration dataset; Bayesian model averaging; BMA-ET; Globe

## 1 Introduction

Terrestrial evapotranspiration (ET) is the physical process by which water is converted from liquid to gaseous state from surface soils, vegetation or water surfaces, and it is accompanied by energy conversion processes (Wang and Dickinson,

2012). Specifically, ET consists mainly of soil evapotranspiration, vegetation transpiration, canopy interception, and



evaporation from water or ice surfaces (Lawrence et al., 2007). Globally, about two-thirds of precipitation is evaporated into the atmosphere each year, while more than half of the net solar radiation absorbed by land surfaces is used to evaporate surface moisture (Jasechko et al., 2013). Evapotranspiration is the nexus of the coupled global water-carbon-energy cycle and is a key process for quantifying the response of the hydrological cycle to global change (Zhang et al., 2024). ET datasets

are classified into four main types in this study: remote sensing–based, machine learning–based, reanalysis–based, and land–surface–model–based (Table 1). These four types of ET datasets are now widely used in scientific research.

**Table 1. Descriptions of four types of evapotranspiration datasets.**

| Dataset basis | Description | Shortcomings |
|---|---|---|
| Remote sensing | Remote sensing models are used to calculate evapotranspiration from the surface by obtaining parameters such as surface vegetation cover, leaf area index, and surface temperature and combining them with meteorological data. | Optical sensors are affected by weather during the acquisition of remote sensing datasets (Schaepman-Strub et al., 2006). |
| Machine learning | Using artificial intelligence–based algorithms in combination with large quantities of meteorological data and multi-source evapotranspiration datasets, evapotranspiration datasets are obtained by calibrating simulated values with measured values (global flux data, ground-based observations, etc.) through rate-setting algorithmic parameters. This method has strong self-learning, self-organisation, and self-adaptation capabilities and is suitable for solving nonlinear and complex problems. | The application of machine learning models in estimating evapotranspiration is somewhat limited, because a priori knowledge of the field still needs to be accumulated due to the complexity of evapotranspiration (Bellido-Jiménez et al., 2021). |
| Reanalysis | Reanalysed information relies on the strengths of numerical weather modelling and assimilation techniques to break through the limitations of raw observational data and produce long series of dynamically and physically coordinated gridded data sets. | Reanalysis data rely on global observations for assimilation, but the coverage and quality of observations vary regionally, with large differences in data quality from one region to another (Parker, 2016). |
| Land surface model | A land surface process model (LSPM) is a mathematical model used to simulate and study energy and moisture exchange at the land surface. It is usually based on physical principles and empirical relationships, dividing the land | Land surface models obtain evapotranspiration data with high temporal resolution but low spatial resolution; model parametric |





| surface into different components (such as soil, vegetation, snow, etc.) and taking into account their interactions; it outputs evapotranspiration by simulating the energy and moisture exchange between the land surface and the atmosphere. | schemes for inhomogeneous subsurfaces are difficult to determine; and there are challenges in terms of multiple observations, data assimilation, etc. (Overgaard et al., 2006). |
| --- | --- |

Against the background of global warming and intensification of the hydrological cycle, global ET is showing obvious
interannual and seasonal variations (Wang and Dickinson, 2012). Therefore, accurate estimation of terrestrial ET is not only necessary for understanding the distribution of water and energy at different spatial and temporal scales but also important for research in many disciplines, such as hydrology and meteorology. However, due to the complexity of spatial and temporal variations in ET, the many factors affecting ET, and the limited availability of field observations, there are large discrepancies among ET estimates from different methods, resulting in large uncertainties in existing assessments of spatial
and temporal variations in global ET (Long et al., 2014).

An important part of hydrological cycle research is ET dataset assessment, which involves comparing and analysing different ET datasets to improve the accuracy, reliability and applicability of ET estimates. In recent years, with the development of remote sensing technology and data processing methods, research on the assessment of ET datasets has
made significant progress. Chao et al. (2021) evaluated five remotely sensed datasets in the continental U.S. using reconstructed ET data (Recon) as a baseline, and the results showed that the GLEAM dataset, P-LSH dataset, and PML dataset showed better performance at the annual scale during 2003–2008, followed by the MTE and MODIS datasets. Yu et al. (2023) validated and assessed the consistency and uncertainty of nine ET datasets in the continental U.S. under normal and extreme weather conditions using eddy covariance (EC) flux observations and the three-cornered hat (TCH) method, and
the results showed that the accuracy of the different ET datasets varied depending on the land cover type and that GLEAM3.6b and two fusion datasets—reliability ensemble averaging (REA) and Synthesized (Syn)—showed superior results for most land cover types during 2003–2015. Pan et al. (2020) analysed the spatial and temporal variations of global terrestrial ET, and the results showed that the annual mean global terrestrial ET ensembles averaged by three different methods were basically the same; results also showed that the ensembles of the remote sensing–based physical models and
the machine learning algorithms both exhibited an increasing trend for ET during the period 1982–2011, while the ensembles of the land surface models on average did not show a significant change. Xie et al. (2024) evaluated the accuracy and uncertainty of seven satellite-based and two reanalysis datasets for global terrestrial ET and found no single ET dataset could provide the most accurate ET estimates for all land cover types; GLASS-AVHRR and GLASS-MODIS were the two datasets with the lowest relative uncertainty, while MERRA2 had the largest relative uncertainty from 2003 to 2015.




To reduce ET data uncertainty, improve data precision, and extend the study period, researchers have used data fusion techniques—such as arithmetic averaging, weighted averaging, and machine learning methods—to integrate multiple ET datasets into a more reliable dataset. Zhang et al. (2019) developed the first set of coupled global terrestrial ET and vegetation gross primary productivity datasets at 500 m resolution for the period 2000–2020 based on the multi-scale fusion

of the PML-V2 ET model with station and basin observations and global remote sensing data. He et al. (2020) developed a gridded ET dataset (1982–2011) from statistical or empirical data, remotely sensed data, and land-surface models for the contiguous United States (CONUS), which was integrated via Bayesian three-cornered hat (BTCH) and ensemble mean (EM) methods, and the results show that BTCH outperforms EM and all individual parent datasets compared to eddy covariance towers ($ET_{EC}$) from AmeriFlux sites and ET values from the water balance method ($ET_{WB}$). Lu et al. (2021) obtained a long

series (1980–2017) of global daily evapotranspiration datasets (REA-ET) with a spatial resolution of 0.25° using the fusion of reliability ensemble averaging methods based on three widely used model-based ET datasets, including ERA5, MERRA2, and GLDAS2-Noah reanalysis data, and the results showed that the fusion dataset captured the evapotranspiration trends in different areas well and performed well for all vegetation cover types.

With the development of remote sensing technology and improved data fusion methods, a large number of ET datasets have emerged. However, existing efforts to fuse ET datasets have not generally involved multiple types of ET datasets but have mainly focused on fusion of a single type of ET dataset, which limits their ability to capture characteristics of regional ET variations. This limitation seriously hampers the accurate quantification of the terrestrial water balance and understanding of climate change impacts. The current study looks at more datasets, including four types of ET datasets, comprising 30 ET

datasets in all, and the ET datasets are also evaluated from more diverse perspectives, from the site scale to the basin scale to the global scale, which can provide data references for studies at different spatial scales. Therefore, this study attempts to develop a long time series (1980–2020) global terrestrial evapotranspiration dataset based on ET from FLUXNET and four types of global evapotranspiration datasets (a total of 30 ET datasets) using a Bayesian model averaging (BMA) algorithm; the aim is to obtain an ET fusion dataset that contains more diverse information than previous efforts. This fusion dataset

features dynamic weighting schemes for different vegetation types based on ET from FLUXNET, as well as non-common coverage years among ET datasets.





## 2 Introduction

### 2.1 Data

**2.1.1 Evapotranspiration data**

Thirty evapotranspiration datasets were used in this study, including four remote sensing (RS) datasets, four machine learning (ML) datasets, two reanalysis (RA) datasets, and 20 datasets based on land surface models (LSMs) (Table 2 and Fig. 1). Download links for all ET datasets are in Table S1 of the supplementary material. The Penman-Monteith-Leuning (PML) estimates are calculated using the observation-driven PML model (Zhang et al., 2016). The basic principle of the Global

Land Evaporation Amsterdam Model (GLEAM) is to maximise the recovery of ET information contained in current satellite-observed climate and environmental variables to derive values for the different components of terrestrial ET (Martens et al., 2017). The Global Land Surface Satellite (GLASS) evapotranspiration dataset is a long-time-series, high-resolution, high-precision remote sensing inversion dataset based on multi-source remote sensing data and measured station data obtained through inversion algorithms (Liang et al., 2013). The PLSH dataset was obtained based on AVHRR remotely

sensed data and reanalysed meteorological data inverted with the improved Penman-Monteith formula (Zhang et al., 2015). The FLUXCOM datasets use meteorological observations and satellite data as inputs to upscale evapotranspiration estimates from 224 flux tower sites using three machine learning methods (random forest algorithm, artificial neural network, and multivariate adaptive regression spline) (Jung et al., 2019). Three FLUXCOM datasets were included in this study—FLUXCOM-CRUNCEP_v8, FLUXCOM-GSWP3, and FLUXCOM-WFDEI—each with different input data. The Model

Tree Ensemble Evapotranspiration (MTE) dataset is based on global flux site observations, combining MODIS data and meteorological data, and using machine learning to extrapolate the data from the site scale to the global scale (Jung et al., 2011). ERA5-Land (the land component of the European Centre for Medium-Range Weather Forecasts [ECMWF] ReAnalysis v5) is produced by inversion of the terrestrial component of the ERA5 climate reanalysis data (Muñoz-Sabater et al., 2021). The production combines climate model data with monitoring data from meteorological observatories to form a

globally complete and continuous data set using the laws of physics. The MERRA-Land (Modern-Era Retrospective Analysis for Research and Applications, land only) dataset focuses more on historical analyses of the hydrological cycle over a wide range of weather and climate scales (Reichle et al., 2011). GLDAS stands for Global Land Data Assimilation System, and its goal is to use advanced land surface modelling, assimilation techniques, and the assimilation of satellite- and ground-based observational datasets to produce optimal land surface states and flux fields. The study used three GLDAS datasets

that differed based on the surface model used: GLDAS CLSM, GLDAS NOAH, and GLDAS VIC (Rodell et al., 2004). Version 12 of the TRENDY programme (TRENDY v12) was chosen for this study (corresponding to dataset IDs 14–30 in Table 2), and all simulations follow the same boundary conditions and simulation specifications and are therefore well comparable (Friedlingstein et al., 2023).

**Table 2. Information for 30 evapotranspiration datasets.**





| ID | Name | Type | Spatial resolution | Time resolution | Period | References |
|----|------|------|--------------------|-----------------|--------|------------|
| 1 | PML | RS | 0.5°×0.5° | mon | 1981–2012 | Zhang et al. (2016) |
| 2 | GLEAM 3.6a | RS | 0.25°×0.25° | mon | 1980–2021 | Martens et al. (2017) |
| 3 | GLASS | RS | 0.05°×0.05° | 8 day | 1982–2018 | Liang et al. (2013) |
| 4 | PLSH | RS | 0.083°×0.083° | mon | 1982–2013 | Zhang et al. (2015) |
| 5 | FLUXCOM-CRUNCEP_v8 | ML | 0.5°×0.5° | mon | 1950–2016 | Jung et al. (2019) |
| 6 | FLUXCOM-GSWP3 | ML | 0.5°×0.5° | mon | 1950–2014 | Jung et al. (2019) |
| 7 | FLUXCOM-WFDEI | ML | 0.5°×0.5° | mon | 1979–2013 | Jung et al. (2019) |
| 8 | MTE | ML | 0.5°×0.5° | mon | 1982–2011 | Jung et al. (2011) |
| 9 | ERA5-Land | RA | 0.1°×0.1° | mon | 1950–2023 | Muñoz-Sabater et al. (2021) |
| 10 | MERRA-Land | RA | 0.5°×0.678° | mon | 1980–2015 | Reichle et al. (2011) |
| 11 | GLDAS CLSM 2.0 | LSM | 1°×1° | mon | 1948–2014 | Rodell et al. (2004) |
| 12 | GLDAS NOAH 2.0 | LSM | 1°×1° | mon | 1948–2014 | Rodell et al. (2004) |
| 13 | GLDAS VIC 2.0 | LSM | 1°×1° | mon | 1948–2014 | Rodell et al. (2004) |
| 14 | CABLE-POP | LSM | 1°×1° | mon | 1700–2022 | Haverd et al. (2018) |
| 15 | CLASSIC | LSM | 1°×1° | mon | 1701–2022 | Melton et al. (2020) |
| 16 | CLM5.0 | LSM | 1.25°×0.9375° | mon | 1701–2022 | Lawrence et al. (2019) |
| 17 | DLEM | LSM | 0.5°×0.5° | mon | 1700–2022 | Tian et al. (2015) |
| 18 | E3SM | LSM | 1.25°×0.9375° | mon | 1700–2022 | Bisht et al. (2018) |
| 19 | EDv3 | LSM | 0.5°×0.5° | mon | 1700–2022 | Ma et al. (2022) |
| 20 | IBIS | LSM | 0.5°×0.5° | mon | 1700–2022 | Yuan et al. (2014) |
| 21 | ISBA-CTRIP | LSM | 1°×1° | mon | 1700–2022 | Delire et al. (2020) |
| 22 | JSBACH | LSM | 1.875°×1.875° | mon | 1700–2022 | Reick et al. (2021) |
| 23 | LPJ-GUESS | LSM | 0.5°×0.5° | mon | 1700–2022 | Smith et al. (2014) |
| 24 | LPJmL | LSM | 0.5°×0.5° | mon | 1700–2022 | Schaphoff et al. (2018) |
| 25 | LPX-Bern | LSM | 0.5°×0.5° | mon | 1700–2022 | Lienert et al. (2018) |
| 26 | OCN | LSM | 1°×1° | mon | 1700–2022 | Zaehle et al. (2010) |





| 27 | ORCHIDEE | LSM | 0.5°×0.5° | mon | 1700–2022 | Vuichard et al. (2019) |
| 28 | SDGVM | LSM | 1°×1° | mon | 1700–2022 | Woodward et al. (2004) |
| 29 | VISIT | LSM | 0.5°×0.5° | mon | 1860–2022 | Ito et al. (2012) |
| 30 | YIBs | LSM | 1°×1° | mon | 1700–2022 | Yue et al. (2015) |

Note: In the table, RS is used for remote sensing, ML for machine learning, RA for reanalysis, and LSM for land surface

model; "mon" indicates a time resolution of one month.

**Figure 1: Years of coverage for each evapotranspiration dataset, with 1982–2011 being the common period of coverage for all evapotranspiration datasets. Dataset types are labelled as follows, by group: RS = remote sensing; ML = machine learning; RA = reanalysis; LSM = land surface model.**



### 2.1.2 FLUXNET2015 flux sites

In order to evaluate the evapotranspiration datasets, the study used observations from FLUXNET2015 flux sites (Fig. S1, Table S2–S4). Flux site data are from the FLUXNET2015 dataset, which covers 212 flux sites worldwide and documents flux observations on near-surface carbon, water, and energy; quality control information; and uncertainty quantification results collected from 1991 to 2014 (accessed at https://fluxnet.org/data/fluxnet2015-dataset). The FLUXNET2015 dataset provides continuous measurements of carbon and energy fluxes at half-hourly to hourly intervals via the eddy covariance method (Baldocchi et al., 2001; Ma et al., 2021). The study subsequently refers to the observational dataset as ET from FLUXNET. Based on MODIS IGBP (International Geosphere-Biosphere Programme) land cover, the flux sites can be classified into 12 vegetation types (see Fig. S1). These sites are distributed across the major climatic zones (tropical, subtropical, temperate, and boreal) and continents (Asia, Africa, South America, Oceania, and especially Europe and North America) (Fig. S1). Specific flux site data processing is described in Supplementary information Text S1. Ultimately, our study retained 174 flux sites. In addition, we also validated the energy balance closure on the flux site data (Supplementary information Text S2 and Fig. S2). We found that the flux sites for most vegetation types were consistent with energy balance closure, so the 174 sites screened by the study are reliable and can be used for accuracy evaluation of ET datasets.

### 2.1.3 Other datasets

To assess global ET datasets at the basin scale, a total of 32 basins across the globe were selected for this study, and basin boundary data were obtained from HydroBasins Level 3 basin boundaries (accessed at https://www.hydrosheds.org/products/hydrobasins#downloads) (Table S5). The streamflow data were obtained from the Dai and Trenberth Global River Flow and Continental Discharge Dataset (http://www.cgd.ucar.edu/cas/catalog/surface/dai-runoff/index.html) and the Global Runoff Data Centre (http://www.bafg.de/GRDC/EN/Home/homepage_node.html) (Table S5). Precipitation data were obtained from the Global Precipitation Climatology Centre (GPCC), which provides a global land-surface gridded precipitation dataset based on approximately 86,100 rain gauges and stations worldwide. The advantage of GPCC is the use of a large number of station observations. The spatial resolution is 1°×1°. Data are available from https://psl.noaa.gov/data/gridded/data.gpcc.html.

### 2.2 Method

Since the common coverage period for the 30 ET datasets is 1982–2011, the ET evaluations in this study are also focused on this time period. ET dataset uncertainty is characterised using standard deviation. The evaluation of ET datasets covers three specific aspects. First, site scale: thirty ET datasets were assessed for accuracy at site scale using the three statistical indicators described below. Evaluations of site scale were carried out based on 1991–2011 (the years of common coverage of the flux station data and the 30 ET datasets).

(1) Correlation coefficient (R)



$$R = \frac{\sum_{i=1}^{n}(x_i - \bar{x})(y_i - \bar{y})}{\sqrt{\sum_{i=1}^{n}(x_i - \bar{x})^2 \sum_{i=1}^{n}(y_i - \bar{y})^2}}, \tag{1}$$

(2) Mean absolute error (MAE)

$$MAE = \frac{\sum_{i=1}^{n}|x_i - y_i|}{n}, \tag{2}$$

(3) Root-mean-square error (RMSE)

$$RMSE = \sqrt{\frac{\sum_{i=1}^{n}(x_i - y_i)^2}{n}}, \tag{3}$$

where $n$ is the number of samples, $x_i$ is the estimated monthly value of the evapotranspiration dataset, $y_i$ is the observed monthly value of the evapotranspiration site, and $\bar{x}$ and $\bar{y}$ are the mean values of $x_i$ and $y_i$, respectively.

Second, the basin water balance approach was used for ET evaluation at the basin scale (Supplementary information Text S3). Third, the global-scale ET evaluation used the three-cornered hat (TCH) method (Supplementary information Text S4). Using the TCH method, we obtained the TCH uncertainty and relative uncertainty for each ET dataset, where the relative uncertainty is calculated as the ratio of the uncertainty to the mean value of the time series. We also used Taylor diagrams for comprehensive evaluation of ET datasets.


Finally, evapotranspiration fusion was completed using a BMA method. The specific BMA fusion steps are shown in Fig. 2 and are explained as follows: (1) sites were partitioned according to land cover type; taking cropland (CRO) as an example, a total of 16 CRO sites were involved, and 60% of the sites (10 sites) were selected to participate in the BMA fusion, while 40% of the sites (6 sites) were used for validation. (2) Due to the overall short coverage time period of the flux sites, all cropland

sites were considered as one site in this study, and the time series were spliced to obtain a longer time series of site observations ($ET_{obs}$). (3) For each ET dataset, data from the 10 site locations and times identified above were spliced, and this step traversed all the ET datasets, resulting in 30 time series ($ET_1, \ldots, ET_{30}$), with length the same as $ET_{obs}$. (4) $ET_{obs}$ was used as an observation to perform a BMA analysis of the 30 ET sequences, thus obtaining the weight of each ET dataset under cropland vegetation type, denoted as $w_{CRO-1}, \ldots, w_{CRO-30}$, where $w_{CRO-1} + \cdots + w_{CRO-30} = 1$. This weight was

applied to all cropland grid points for each ET dataset. (5) Fusion of all ET datasets was achieved using the following: $ET_{CRO-merge} = ET_{CRO-1} \times w_{CRO-1} + \cdots + ET_{CRO-30} \times w_{CRO-30}$. (6) The above steps 1 to 5 were repeated to obtain the ET fusion datasets for all other vegetation types. (7) The ET fusion datasets for each vegetation type were spliced to obtain the final global ET fusion dataset. (8) The remaining six cropland stations from step 1 were used to validate and evaluate the accuracy of the global ET fusion datasets in cropland. The BMA analyses were carried out based on 1991–2011 (the years of



common coverage for the flux station data and the 30 ET datasets). The datasets obtained from the final fusion were denoted as BMA-ET. It is important to note that land cover types change over time, and 2001 land cover classifications are used in this study. Also, note that the 12 vegetation cover types do not cover the entire study area. For areas not covered, an equal weighting approach was taken. Calculation formulas for the BMA algorithm are shown in Supplementary information Text S5. The weighting scheme for determining non-common coverage years in the ET fusion process is addressed in

Supplementary material Text S6 and Fig. S3.

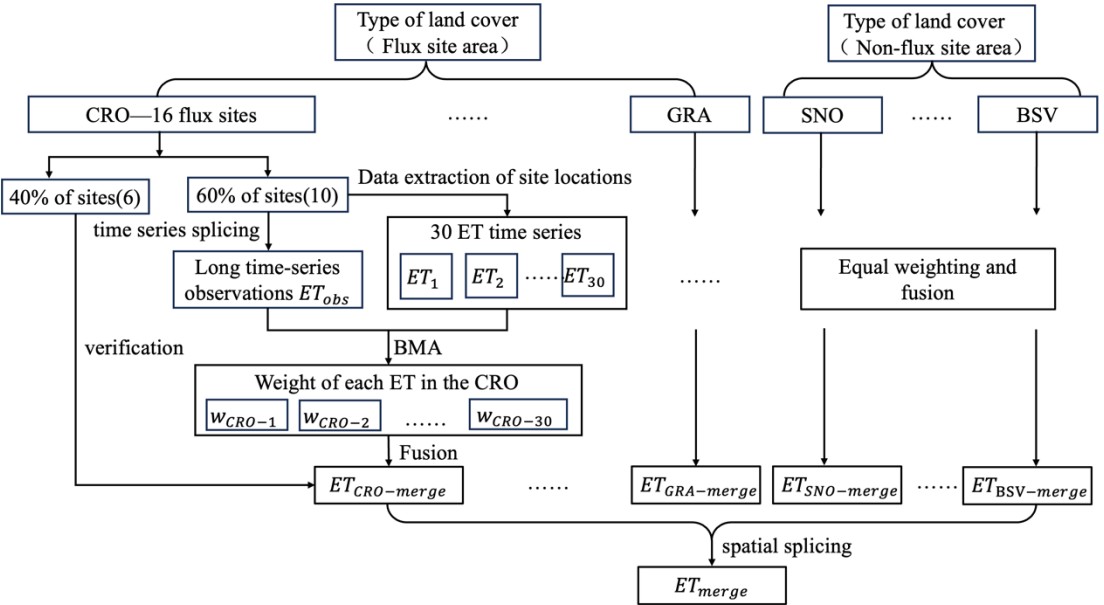

**Figure 2: Steps for fusion of ET datasets based on the BMA algorithm.**

## 3 Results

### 3.1 Evaluation of evapotranspiration data

The basic spatial pattern found for evapotranspiration is that it is larger at low equatorial latitudes and smaller at high latitudes and altitudes (Fig. S4 and S5). The global spatial patterns across the 30 evapotranspiration datasets are generally similar, but some differences are apparent (Fig. S4). In the Amazon and Central African regions, evapotranspiration is higher for GLASS, JSBACH, ISBA-CTRIP, and SDGVM and lower for GLDAS-VIC, CLASSIC, and LPX-Bern. In terms of evapotranspiration categories, remote sensing–based ET datasets are larger and machine learning–based ET datasets are

smaller in the above regions (Fig. S5). Overall, ET shows a three-peaked distribution of equatorial high and bipolar low, globally. Remote sensing–based evapotranspiration datasets have the largest evapotranspiration near the equator, with mean values of evapotranspiration exceeding 1300 mm in this latitudinal band (Fig. S5). In particular, the greater the multi-year average of global evapotranspiration, the greater the differences among ET datasets (Fig. S6). After considering ET values,

ET uncertainty (variation among ET datasets) is greater in the Sahara Desert, the Middle East, the Tibetan Plateau, and
Central Australia (Fig. S7). The interannual trends in ET datasets also show large differences (Fig. S7). The largest trends in
ET are in ERA5-Land and MERRA-Land, where the decreasing trends in ET exceed 4 mm·yr⁻¹ at low latitudes. ET trends
of the three FLUXCOM datasets and MTE are small in most parts of the globe, with both increasing and decreasing trends of
less than 1 mm·yr⁻¹. Overall, machine learning–based ET datasets show a smaller trend during 1982–2011, while reanalysis-
based ET datasets have a larger decreasing trend at low equatorial latitudes (Fig. 3). ET shows an increasing trend in regions
such as northern North America, Southern Africa, North Asia, and Australia.

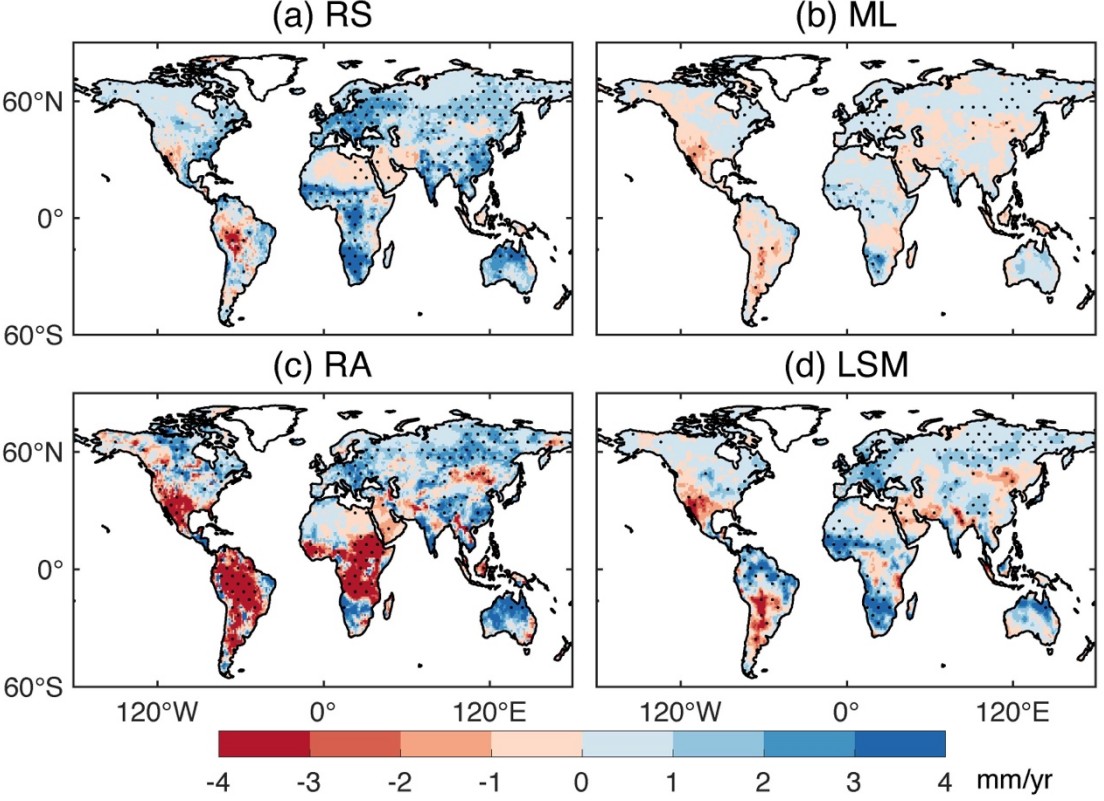

**Figure 3: Interannual trends of multi-dataset medians in the four categories of evapotranspiration datasets during 1982–2011 (the black dotted areas indicate that the trend in these cells was significant at the 95% level).**

### 3.1.1 Site scale

Using data from the ET from FLUXNET, we evaluated the accuracy of the 30 ET datasets over the globe at site scale. The
correlation coefficients for the four types of ET data are similar, but the MAE and RMSE are lower for the land-surface-
model-based ET datasets (Fig. 4). The correlation coefficients for PML, GLASS, and MTE data are the highest among the
30 ET datasets, while the MAE and RMSE are the lowest for GLADS-VIC, EDv3, LPJmL, and LPX-Bern. Over 60% of
sites in most of the ET datasets have correlation coefficients of 0.6 or higher, with higher correlation coefficients and lower
MAE (less than 10 mm) and RMSE (less than 20 mm) in regions such as Europe and North America (Fig. S8, S9 and S10).



These are also the areas with the highest density of flux sites. On a seasonal scale, MTE performs better in all four seasons, while remote sensing–based ET datasets, especially the PML dataset, perform superiorly in March–May (MAM) and June–August (JJA) (Fig. S11, S12 and S13). GLDAS-VIC also performs better in JJA. LPJmL and machine learning–based ET datasets perform better in September–November (SON) and December–February (DJF) (Fig. S11, S12 and S13). In addition,

CABLE-POP performs better in SON and GLASS performs better in DJF. In terms of vegetation types, MTE performs optimally in most vegetation types, except SAV, WET, and WSA (Fig. S14, S15 and S16). PML excels in vegetation types other than CRO, MF, and WSA. For EBF, GRA, and SAV, machine learning–based ET datasets perform better; while for other vegetation types, remote sensing–based ET datasets perform better. Of these, for CRO, ET datasets based on both remote sensing and land surface process models perform better. Each ET dataset performs poorly in both SAV and WSA

vegetation, with most ET datasets having correlation coefficients of less than 0.2 (Fig. S17). Most of the ET datasets perform well in CSH, GRA, and MF, and the remote sensing–based ET datasets perform extremely well in CSH, with correlation coefficients of 0.8 or more. After considering the performance of each ET dataset in terms of season and vegetation type, we recommend remote sensing– and machine learning–based ET datasets (Table 3).





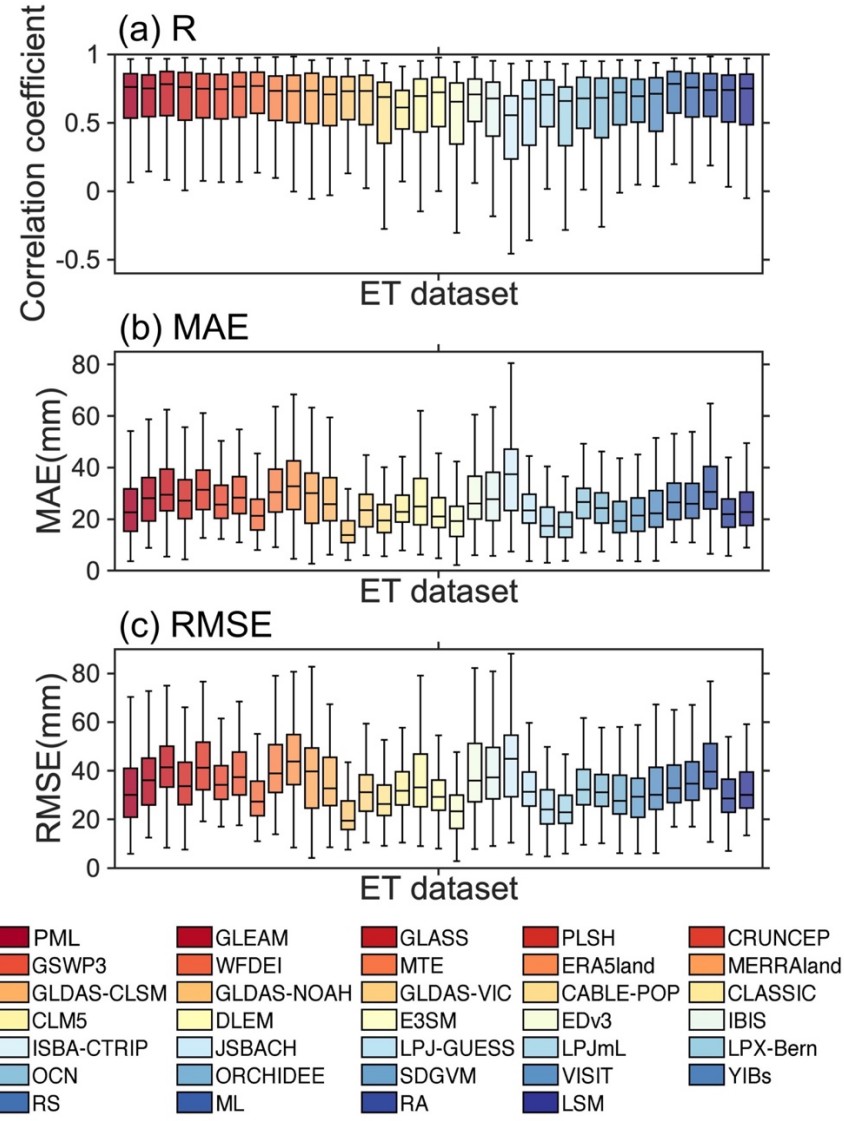

**Figure 4: Correlation coefficients (R), mean absolute error (MAE), and root-mean-square error (RMSE) for ET from FLUXNET and each ET dataset during 1991–2011 (for each box, the centre horizontal line represents the median of the sample, the upper and lower borders of the line represent the 75th and 25th percentiles, respectively, and the two 'whiskers' extend to the maximum and minimum values that are not considered outliers; outliers are plotted separately).**

**Table 3. Guidelines for the use of ET datasets.**

| Classification | | Recommendation | |
| --- | --- | --- | --- |
| Level 1 | Level 2 | ET datasets | ET type |
| Season | MAM | MTE; PML | RS |
| | JJA | MTE; GLDAS-VIC; PML | RS |





|  |  |  |  |
|---|---|---|---|
|  | SON | MTE; LPJmL; CABLE-POP | ML |
|  | DJF | MTE; GLASS; LPJmL | ML |
|  | CRO | MTE; PLSH; SDGVM | RS; LSM |
|  | DBF | MTE; PML; SDGVM | RS |
|  | EBF | MTE; CABLE-POP; PML | ML |
|  | ENF | MTE; PML; CABLE-POP | RS |
| Vegetation | GRA | MTE; GLDAS-VIC; PML | ML |
|  | MF | MTE; VISIT; PLSH | RS |
|  | OSH | PML; MTE; PLSH | RS |
|  | SAV | PML; WFDEI; CRUNCEP | ML |
|  | WET | PML; GLDAS-VIC; GLDAS-NOAH | RS |
|  | WSA | GLASS; PLSH; CLM5 | RS |

### 3.1.2 Basin scale

By comparing the multi-year average evapotranspiration from the 30 ET datasets with the multi-year average observed evapotranspiration from the basin water balance, we found that the $R^2$ of the two reached more than 0.8 in most of the ET datasets, except for EDv3 (Fig. S18). In nearly two-thirds of the ET datasets, $R^2$ actually exceeds 0.9. Compared with water balance–based observational ET, ISBA-CTRIP, JSBACH, and GLDAS-CLSM perform better at the basin scale compared to other ET datasets (Fig. S19). Among them, only EDv3 and GLASS have correlation coefficients below 0.9. Remote sensing– and machine learning–based ET datasets have relatively greater TCH uncertainty, especially for PML, GLASS, CRUNCEP, and MTE, and TCH uncertainty is generally greater in low-latitude basins than that in high-latitude basins (Fig. S20). However, in terms of relative uncertainty for TCH, the relative uncertainty is greater for CRUNCEP, MERRA-Land, and MTE (Fig. 5). At the same time, the relative uncertainty of TCH in the low- and high-latitude basins shows the opposite behaviour of the TCH uncertainty, being smaller in the low-latitude basins and larger in the high-latitude basins (Fig. 5). Land-surface-model-based ET datasets have less relative uncertainty of TCH at the basin scale compared to the other three types of ET datasets, whereas machine learning– and reanalysis-based datasets have greater TCH relative uncertainty.



**Figure 5: Spatial distribution of relative uncertainty of TCH during 1982–2011 for 32 basins (%).**

**3.1.3 Global scale**

The machine learning–based dataset MTE-ET is chosen here as the reference data. The correlation coefficients of remote sensing–based, machine learning–based, and land-surface-model-based ET datasets are all around 0.7 at the global scale,





with machine learning–based ET datasets having the smallest RMSE, and land-surface-model-based and remote sensing–based ET datasets having the second smallest RMSE (Fig. 6). Reanalysis-based ET datasets are the worst performers

globally, with the lowest correlation coefficient and the largest RMSE. The TCH analysis of global ET datasets shows that machine learning–based ET datasets have relatively low uncertainty, especially MTE (Fig. S21). The MTE dataset has less uncertainty at high latitudes and more uncertainty at low latitudes, while the relative uncertainty is the opposite. Uncertainties in the datasets MERRA-Land, CLM5, EDv3 and JSBACH are large, whereas uncertainties in GLDAS-CLSM and SDGVM are smaller in the northern hemisphere and larger in the southern hemisphere. For TCH relative uncertainty,

machine learning– and remote sensing–based datasets have less relative uncertainty. (Fig. S22). GLDAS-VIC, CLM5, EDv3, and LPX-Bern have large relative uncertainties of TCH, and areas with larger relative uncertainties are predominantly located at high latitudes. Overall, the relative uncertainty is lower in regions with larger ET values, such as the Amazon, Central and Southern Africa, and Southeast Asia.

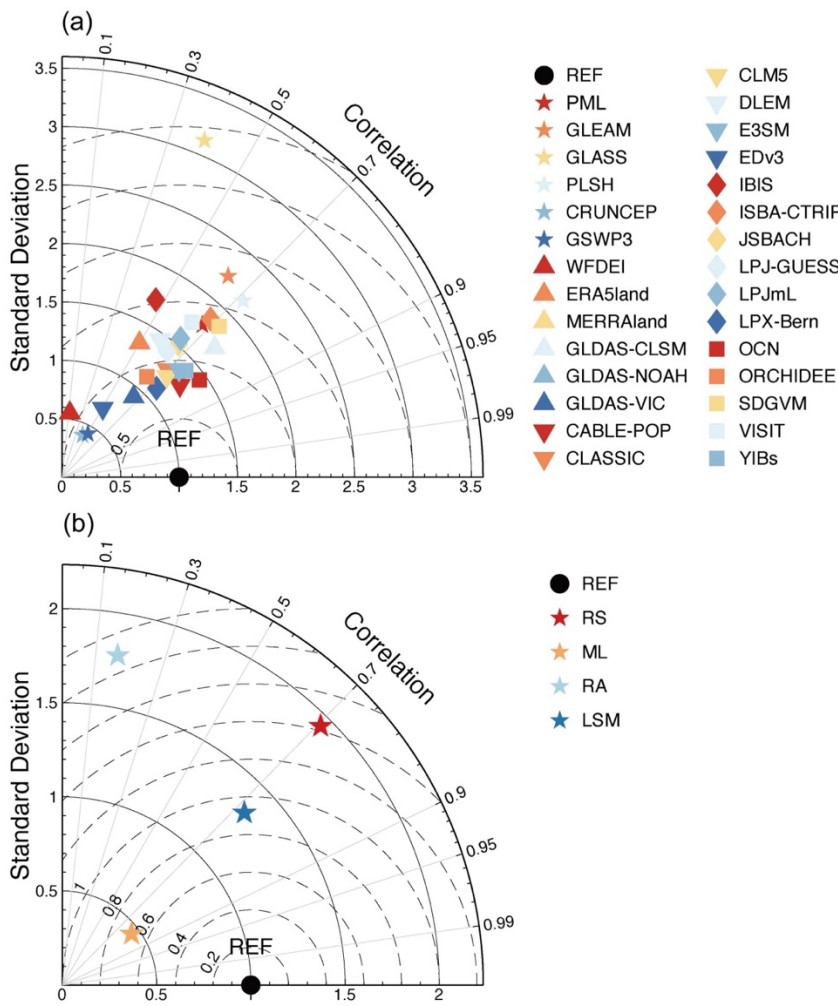



**Figure 6: Taylor diagram of annual ET datasets during 1982–2011 (a) for 30 ET datasets, (b) for four types of ET datasets (MTE-ET is chosen here as the reference data).**

### 3.2 Evapotranspiration dataset fusion

Thirty global ET datasets were fused based on the BMA approach, hereafter referred to as BMA-ET (Fig. 7). Areas of high evapotranspiration are mainly found at low equatorial latitudes, such as the Amazon, central Africa, and Southeast Asia. The

Sahara Desert, the Middle East, and the Tibetan Plateau have less evapotranspiration. ET trends are positive in most parts of the globe during 1980–2020, and the main regions with high ET trends are the Amazon, Europe, Southern Africa, and the Indian Peninsula. Decreasing trends in ET are apparent in a few regions, such as southern North America, southern South America, eastern Africa, western Asia, and northern India, with some decreasing trends of more than 3 mm·yr$^{-1}$ (Fig. 7). From 1980 to 2020, the global terrestrial averaged ET showed an overall increasing trend (0.21 mm·yr$^{-1}$, $p<0.01$) (Fig. S23).

At the same time, the variation in ET is highly seasonal, with ET exceeding 50 mm in July, while in January and December ET is around 30 mm (Fig. S23).

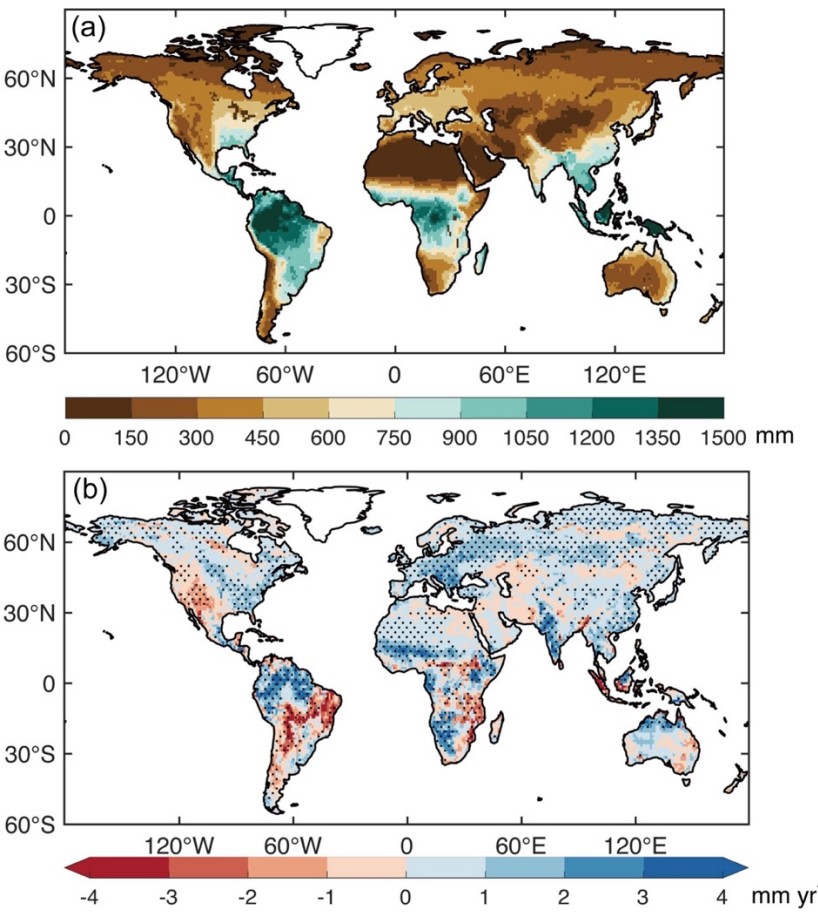



**Figure 7: Spatial distribution of multi-year averages and trends in BMA-ET during 1980–2020. The black dotted areas indicate the trend passes the significance test (*p*<0.05) at that grid point.**

We have also verified the accuracy of the new fusion dataset, BMA-ET. Validation using flux tower ET as reference data shows that more than 70% of the flux sites have correlation coefficients higher than 0.8 and that the flux sites with high correlation coefficients are mainly located in the mid-to-high latitudes of the northern hemisphere (Fig. S24). Similarly, flux sites with low MAE and RMSE are mainly located in the above areas. The results show that BMA-ET has higher correlation coefficients and lower root-mean-square errors than most of the individual ET datasets (Fig. 8). BMA-ET also performs well

in most vegetation types, except for evergreen broadleaf forest (EBF), which has poor performance in all ET datasets (Fig. S25). The correlation coefficients of BMA-ET exceed 0.7 in most vegetation types. At the basin scale, the performance of BMA-ET is also outstanding (Fig. S26). The correlation coefficient of BMA-ET is more than 0.95 and the RMSE is less than 0.4.

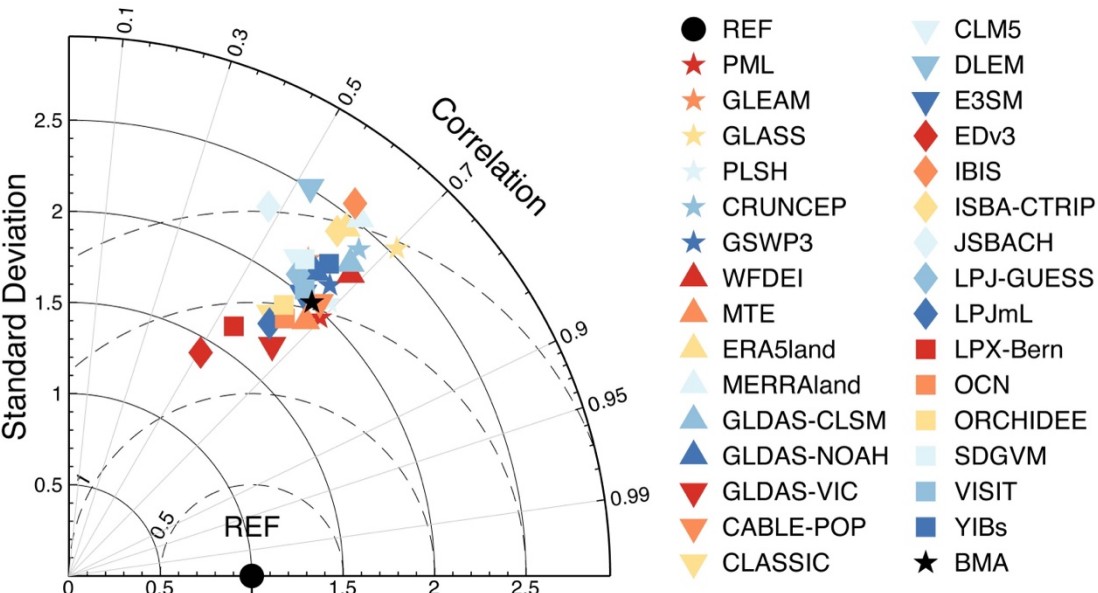

**Figure 8: Taylor diagram of ET datasets at all stations from 1991 to 2011. The observation data are ET from FLUXNET.**

## 4 Discussion

### 4.1 Uncertainty in ET from FLUXNET

ET from FLUXNET are often used as reference data to validate and assess the estimation accuracy of different ET datasets, but turbo-correlation methods based on EC instruments suffer from energy non-closure (Foken, 2008; Foken et al., 2006),

whereby the residual difference between net radiative and soil heat fluxes is always higher than the sum of sensible and latent heat fluxes (Wilson et al., 2002). According to Twine et al. (2000), this unclosed energy can be as high as 10% to 30%,



and the latent heat fluxes in ET from FLUXNET used in this study were corrected for the energy non-closure using the MDS method, but the current method still suffers from the problem of energy imbalance (Mauder et al., 2008). The reasons for this energy underestimation are complex and there is no unanimous conclusion, and researchers agree that the most universal
function needs to be re-established to avoid underestimating both sensible and latent heat fluxes (Finnigan, 2004; Finnigan et al., 2003).

In addition, the use of site observations to assess large-scale evapotranspiration datasets still has some drawbacks, such as spatial scale mismatch and sparse site distribution. Typically, flux sites only cover a spatial range of a few hundred metres
(Soegaard et al., 2003), whereas an image in an ET dataset usually covers a range of several kilometres or even tens of kilometres, which makes a single site unrepresentative of a single image or its neighbouring images. Even though both ET estimates are relatively accurate, the different spatial resolutions can result in large differences in the estimates (Mauser and Schädlich, 1998; Sharma et al., 2016). Due to global climate differences, topographic undulations, and surface heterogeneity, a limited number of stations cannot represent all surface conditions around the globe. The flux sites used in this study are
sparsely distributed globally, with most sites in North America and Europe, and fewer sites on the other continents. The deserts, wetlands, and other areas lack observational data, and the accuracy of their estimation cannot be guaranteed.

**4.2 Limitations of evaluation method**

The water balance method is often used as a standard method for estimating ET in watersheds (Liu et al., 2023; Ma et al., 2024). However, there are limitations and uncertainties in assessing the accuracy of ET datasets using this method. For
example, the method is able to obtain only the average value of ET in a basin but not the spatial distribution of ET in a basin. Total water storage change (TWSC) cannot be ignored at higher temporal resolution because TWSC greatly affects basin ET estimates. Climate change and human activities have significantly altered the regional water cycle over the past few decades, resulting in dramatic changes in total water storage. In addition, the evaluation of basin-scale accuracy depends on the quality of precipitation and runoff data.

Since there are no observations of ET at the global scale for reference, this study directly uses the MTE, an ET dataset obtained based on ET from FLUXNET and machine learning algorithms, as the true value of ET. Cross-validation results have shown that the method predicted site-scale ET with a model efficiency of up to 0.84 (Jung et al., 2011). Since its release, this dataset has been used in many studies on water balance, atmospheric interactions, and validation of land surface model
results (Draper et al., 2018; Xu et al., 2019). However, the disadvantage of machine learning algorithms is that model accuracy is dependent on the data, including issues of data quality, data processing methods, data representativeness, and data scale. MTE datasets have coarse spatial and temporal resolution and are suitable for water and energy cycle studies only at national and global scales, making it difficult to meet the needs for fine quantification of water balance and assessment of water-use efficiency at field scale (Ke et al., 2017).




In addition, the uncertainty quantified by the TCH method does not reflect the goodness of the modelling of the evapotranspiration datasets, but only the magnitude of the uncertainty, while the relative uncertainty, although indicative of the extent to which the different datasets are modelled, can be larger for areas with smaller evapotranspiration due to the order of magnitude of the datasets.

**5 Data availability**

This long-term global terrestrial ET dataset covers the period of 1980–2020. It contains data for spatial resolutions of 0.5° × 0.5° and 1° × 1° covering the domain of –89.75–89.75° N, –180–179.5° E. The NetCDF formatted output files of the BMA-ET dataset are freely accessible at https://doi.org/10.6084/m9.figshare.28034666.v1 (Wu and Miao, 2024).

**6 Conclusions**

In this study, 30 ET datasets of four types are comprehensively evaluated, including at the point scale, basin scale and global scale, and the following conclusions were obtained:

(1) The machine learning–based ET datasets show a smaller trend during 1982–2011, while reanalysis-based ET datasets have a larger decreasing trend at low equatorial latitudes. ET shows an increasing trend in regions such as northern North America, Southern Africa, North Asia, and Australia.

(2) Based on using ET from FLUXNET as references, we recommend remote sensing– and machine learning–based ET datasets. Among them, our study recommends MTE and PML; however, the selection of ET datasets follows the season and vegetation type. At the basin scale, land-surface-model-based ET datasets have less relative uncertainty of TCH compared to other types of ET. At the global scale, the uncertainty is lower in regions with larger ET, such as the Amazon, Central and Southern Africa, and Southeast Asia.

(3) In this study, a long-time-series (1980–2020) ET fusion dataset (BMA-ET) was generated using a Bayesian model averaging (BMA) method. From 1980 to 2020, the global terrestrial BMA-ET shows an overall increasing trend (0.21 mm·yr$^{-1}$) and has strong seasonality. The results show that BMA-ET has higher correlation coefficients and lower root-mean-square errors than most of the individual ET datasets included in this study. Validation using ET from FLUXNET as reference data shows that correlation coefficients for more than 70% of the flux sites exceed 0.8.


Estimating terrestrial evapotranspiration at large scales has always been a challenge for hydrology and ecology because of the heterogeneity of the subsurface and the interaction between the atmosphere and the land surface. In future research, we should focus on the following two factors: (1) Given the limited availability of site observations, we should improve the

spatial resolution of the dataset as much as possible to match the site scale while studying the energy closure problem in depth. Currently, the regional-scale ET estimates have achieved high accuracy by optimising the model parameters, but the global scale still needs to be calibrated to the observed values. Therefore, in future studies, the density of the site distribution and the accuracy of observations will be the main focus for improving the modelling results. (2) It is necessary to focus on the influence of different factors on the estimation of ET under different climate models in order to compare different ET models, to pay attention to the differences between the models, and to optimise or improve the parameters of the models in

order to improve the accuracy of the model estimation.

**Author contributions**

YW and CM contributed to designing the research; YW implemented the research and wrote the original draft; CM supervised the research; all co-authors revised the manuscript and contributed to the writing.

**Competing interests**

The contact author has declared that none of the authors has any competing interests.

**Acknowledgments**

This research was supported by the National Natural Science Foundation of China (No. 42342023), the State Key Laboratory of Earth Surface Processes and Resource Ecology (2022-ZD-03) and the Fundamental Research Funds for the Central Universities. We thank all contributors and also our external cooperation partners for their support and work.

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
