# Peer review of "Multi-spatial scale assessment and multi-dataset fusion of global terrestrial evapotranspiration datasets"

_Earth System Science Data, 2024_

## Author Comment (AC4)

**A DETAILED LIST OF RESPONSES**
**TO REVIEWER #1**

**Anonymous Reviewer #1 comments**

**General comments:**

Reviewer #1: The research entitled "Multi-spatial Scale Assessment and Multi-dataset Fusion of Global Terrestrial Evapotranspiration Datasets" meticulously evaluated the accuracy and uncertainty inherent in thirty ET datasets at multiple spatial scales. These datasets encompass a variety of methodologies, including those derived from remote sensing－based, machine learning－based, reanalysis－based, and land－surface－model－based. Then the study produced a fusion ET dataset (BMA-ET) using BMA method and dynamic weighting scheme for different vegetation types. The article is well-written and demonstrates strong logical coherence. However, I am doubt about the purpose of this study. As the authors have pointed out,  "there are large discrepancies among ET estimates from different methods", I am wondering how does the research handle the uncertainty between different types of ET datasets. Due to differences in algorithm frameworks and input data, the uncertainty of estimation results varies. The ET Fusion not only combines the advantages of different models, but also integrates uncertainty and even enhances errors. Regarding this, the author did not provide a solution. For a global ET dataset, data availability is more important than validation accuracy, and the results and novelty do not reach the desired level, which I do not think meet the requirements of ESSD. Thus, I recommend rejection. Please see my specific comments below.

**Response:** We greatly appreciate your careful reading of the manuscript, insightful comments, and valuable suggestions. Your thoughtful review has enhanced our paper considerably. The manuscript has been revised thoroughly according to your comments and those of the individual reviewers, with our point-by-point responses detailed below.

**Specific comments:**

1. I think the most significant problem with this research is that all the machine learning ET models and some other models (GLASS, PML, etc.) have been calibrated by ground observations from FLUXNET. The BMA-ET generated in this study used FLUXNET observations to fuse thirty ET datasets, which poses a problem of data reuse, and the estimated results may even overfit.

**Response:** We thank the reviewer for your valuable suggestion. Indeed, the use of FLUXNET2015 data for calibration or validation in some datasets has affected the results to some extent. To address the above issues, we completed the following two components.

(1) 30 datasets are clustered and then fused

First, we calculated a Pearson correlation coefficient matrix using the residuals from the 30 sets of ET datasets with observations from FLUXNET2015 sites. Second, the 30 sets of ET datasets were clustered based on the residual correlation coefficient matrix. Third, for each vegetation type, the BMA fusion of the ET data within each cluster was performed first, and then the BMA fusion of the fused data for each cluster was performed.

(2) Independent data source validation

We used independent data sources to validate the fusion dataset BMA-ET, specifically including AmeriFlux, ChinaFlux, and ICOS. The site information of AmeriFlux, ChinaFlux, and ICOS list in Tables S5–S7. Additionally, we also evaluated the accuracy of BMA-ET using FLUXNET2015 data from 2012 to 2015. The results demonstrate that BMA-ET outperforms other external datasets, achieving correlation coefficients of 0.61, 0.72, and 0.74 with site-level ET measurements from AmeriFlux, ChinaFlux, and ICOS, respectively (Figure R1). Using FLUXNET2015 as reference observations, BMA-ET showed a correlation coefficient of 0.58 with FLUXNET2015 site-level ET during 2012–2015 (Figure R1), while also demonstrating high accuracy across various vegetation types (Figure R2).

[Figure]

Figure R1 (Corresponding to Figure 9). Accuracy evaluation of BMA-ET. The observation data is ET from (a) FLUXNET2015 during the period 2012–2015, (b) AmeriFlux during the period 1994–2020, (c) ChinaFlux during the period 2003–2010, (d) ICOS during the period 2003–2010.

[Figure]

Figure R2 (Corresponding to Figure S28). Accuracy evaluation of BMA-ET under different vegetation types during the period 2012–2015. The observation data is ET from FLUXNET2015.

We have made the following additions in section 2.2 and 3.2.

2. How did the authors handle the estimation accuracy of sparse areas such as South America and Africa during the fusion process?

**Response:** Thank you for this pertinent advice. In this study, ET data were fused not by continent, but also by vegetation type. It is important to acknowledge that the number of sites is small in some vegetation types, and because of the overall short time period covered by the flux sites, this study spliced the time series of all the sites in each vegetation type, resulting in a longer time-span data set of site observations to

be used as observations for the BMA analysis. The relevant content has been described in detail in the section 2.2.

3. The BMA is not an advanced fusion algorithm. The GLASS v4.0 integrated five ET algorithms using BMA in 2014 and upgraded to v5.0 using a deep learning algorithm in 2022. Which version of GLASS product was fused in this study? Why don't the authors consider using deep learning fusion algorithms?

**Response:** Thank you for this comment. We considered GLASS v5, but since its temporal coverage is 2001–2015 and the fusion process requires the introduction of an evapotranspiration product with a longer coverage period, GLASS v4 was finally selected for this study. The GLASS v4 evapotranspiration product uses a Bayesian approach to estimate global land surface latent heat fluxes by combining five traditional latent heat flux algorithms (MOD16 algorithm, improved PM, PT-JPL, MS-PT, and Semi-Empirical Penman Algorithm), using observations from 240 flux sites around the globe as a reference to determine the weighting values for each algorithm. In this study, the Bayesian model averaging method was chosen over deep learning algorithms for the fusion of multiple sets of ET products for the following reasons:

(1) Explicit uncertainty quantification and probabilistic outputs

BMA provides probabilistic outputs (e.g., confidence intervals) by directly quantifying the uncertainty of the model weights and the error distribution of the input data through a Bayesian framework (Vrugt et al., 2008). For example, when fusing multi-source ET data, BMA can explicitly give the contribution weights of different data sources and their uncertainties, which facilitates the assessment of the reliability of the fusion results (Vrugt and Robinson, 2007). Deep learning methods typically output deterministic results and require the additional introduction of a probabilistic framework to quantify uncertainty, but are computationally complex and weakly interpretable (Zhang and Zhu, 2018).

(2) Physical Interpretability and Model Transparency

The BMA approach preserves the physical meaning of each input model (e.g. Penman-Monteith equations, remote sensing inversion algorithms), and reflects the applicability of different models under specific conditions through the assignment of weights (Vrugt et al., 2008). The results can be directly related to the physical mechanisms of the input models, compounding the need for mechanism interpretation in geoscientific studies. Deep learning methods, as 'black box' models, have internal feature representations that are disconnected from the physical process, making it difficult to explain the logic of weight assignment or to correct for sources of model bias (Castelvecchi, 2016).

(3) Robustness for small sample data

When the amount of data is limited (e.g., sparse ground validation sites), the BMA method can avoid overfitting through Bayesian prior distributions and model weight optimization, and is particularly suitable for regional-scale ET fusion (Vrugt et al., 2008). For example, when fusing a small amount of vorticity covariance flux data with multi-source ET products, BMA can constrain the range of weights through prior knowledge. Deep learning methods, on the other hand, are prone to overfitting or underfitting when there are insufficient samples (Zhang et al., 2019), and data augmentation strategies are limited by physical plausibility in geomatics.

4. Table 2 shows that the spatial resolutions of the 30 ET datasets are different. How did the author solve the problem of spatial scale mismatch during the fusion process?

**Response:** Indeed, as you mentioned, the 30 ET datasets have different spatial resolutions. Therefore, we used a bilinear interpolation method to unify the spatial resolution of all the ET datasets to 0.5°×0.5° and 1°×1° before performing the fusion of the multi-source ET datasets.

We have made the following additions in section 2.2.

5. The 30 ET datasets cover different time ranges. How to carry out ET fusion for years with missing ET data?

**Response:** We sincerely thank the reviewer for noting this. Since different ET datasets cover different years, for the common coverage years 1982–2011, the weights of each ET dataset under each vegetation type were obtained by performing BMA analyses based on 30 ET datasets, and these weights were applied to all the years 1982–2011. For the non-common coverage years 1980–1981 and 2012–2020, the weights for each year were obtained by filtering all ET datasets covering that year for BMA analysis to obtain the corresponding weights. The BMA analysis process mentioned above is all based on the years 1991–2011, except that the number of ET products involved in the BMA analysis is changing.

We have made the following additions in section Text S6 and Figure S5.

6. What are the spatial and temporal resolution of BMA-ET? How to handle the mismatches with 30 ET input datasets?

**Response:** In order to meet the needs of research at different spatial scales, we produced 2 sets of BMA-ET datasets with spatial resolutions of 0.5°×0.5° and 1°×1°. At the spatial scale, we used bilinear interpolation to unify the spatial resolution of the 30 sets of ET datasets to 0.5°×0.5° and 1°×1°. In turn, the subsequent ET evaluation and fusion were performed. At the temporal scale, the response content of the fifth comment was attended.

We have made the following additions in section 2.2, Text S6 and Figure S5.

7. Is the observation interval of the ground measurements from FLUXNET half an hour? How to process observation data into monthly scale? Is nighttime observation data used?

**Response:** In this study, half-hourly data from FLUXNET2015 were selected as ET site observations. The half-hourly scale ET data were aggregated to the monthly scale,

resulting in monthly scale ET site data for subsequent analyses. Given that flux data from nighttime eddy covariance measurements are usually subject to large deficiencies and errors, only daytime latent heat flux data were selected for this study. Daytime was defined as the period from 07:00 to 19:00 local time. The study did not use incident shortwave radiation to define the daytime period because occasionally large values of shortwave radiation occur at night.

We have made the following additions in section Text S1.

8. In line 181: What do 10 sites refer to? Does it refer to 60% of CRO sites? Please explain more clearly.

**Response:** I'm sorry that we didn't make this clear in the paper. Here, 10 sites refer to 60% of the CRO sites.

We have made the following additions in section 2.2.

9. In section 2.2 (lines 176-195), "The ET fusion datasets for each vegetation type were spliced to obtain the final global ET fusion dataset". How to obtain the boundaries of vegetation types at the regional scale? What is the accuracy? Have authors considered the fusion errors caused by land cover classification errors?

**Response:** Thank you for this pertinent advice. In general, the vegetation cover type boundaries are curves, i.e., Shapefile vector boundaries. However, in this study, the vegetation cover type we use is grid point data. And the vegetation cover type grid point data is produced by the previous study. The MCD12Q1 data are considered in this study. The MCD12Q1 data is a fusion of data from sensors Terra and Aqua, with a spatial resolution of 500 metres, and provides interannual global data on land cover types (from 2001 onwards), containing six classification systems, where the International Geosphere Biosphere Programme (IGBP) is used (Cai et al., 2014). The IGBP classifies land cover types into 17 categories, including 11 natural vegetation classifications, 3 land use and land mosaics, and 3 unvegetated land classifications.

The product uses supervised classification in addition to additional post-processing of the data, i.e., some a priori knowledge and ancillary data are incorporated to improve the accuracy of the classification. This study uses the MOD12 Q1 dataset to classify global land cover. The following maps show the spatial distribution of masks globally (Figure R3) and for each of the 12 land cover types (Figure R4).

[Figure]

Figure R3 (Corresponding to Figure S3). Global spatial distribution of land cover types based on MOD12Q1.

[Figure]

Figure R4. Global spatial distribution of 12 types of land cover types based on MOD12Q1.

On this basis, we also considered the comments of Reviewer #2. We performed a dynamic comparison of the data set MOD12Q1 for the three periods of 2001, 2010 and 2020 (Figure R5). We found that the consistency of the land cover types between the years was high, with the proportions of consistency between 2001 and 2010, 2010 and 2020, and 2001 and 2020 being 0.80, 0.86 and 0.78, respectively (Figure R6). In addition, we also analyzed the percentage of global land area covered by 12 vegetation types (Figure R7). The results show that the proportion of area covered by various vegetation types does not vary much between years, especially for the four main vegetation types, OSH, WSA, SAV and GRA, which account for a larger proportion of the area.

[Figure]

Figure R5 (Corresponding to Figure S33). Spatial distribution of land cover changes in three periods (2001, 2010, 2020) based on MOD12Q1.

[Figure]

Figure R6 (Corresponding to Figure S34). Consistency between MOD12Q1 land cover types across years. Subfigures a-c show the level of consistency between land cover types for 2001 and 2010, 2010 and 2020, and 2001 and 2020, respectively. The level of consistency is characterized by the ratio of the number of grid points with consistent vegetation types to the total number of grid points on land.

[Figure]

Figure R7 (Corresponding to Figure S35). Percentage of global land area covered by 12 vegetation types based on MOD12Q1. The bars in each subfigure represent the proportion of vegetation types in 2001, 2010 and 2020, respectively.

We have made the following additions in section 4.1.

10. In Figure 2, "the 12 vegetation cover types do not cover the entire study area. For areas not covered, an equal weighting approach was taken". Is this weight scheme reasonable?

**Response:** Thank you for your comment. Areas not covered by the ten vegetation cover types could not be analyzed for BMA as no site information was available. Here, we introduce a Bayesian-Three Cornered Hat (BTCH) method. This method is an advanced statistical approach for multi-source data fusion and error estimation, which is particularly applicable to the fields of remote sensing, Earth observation and model evaluation. Its core advantage lies in its ability to estimate the error characteristics (e.g., random error, systematic bias) and their relative weights of multiple data sources simultaneously without relying on real reference data. The computational process of BTCH method is detailed in Text S8.

We compared the results of ET fusion of uncovered areas of 10 vegetation types based on the BTCH method and the equal weighting method (Figure R8). The results showed that the correlation coefficients between the ET estimates based on the equal weighting method and the BTCH-based method were high in the area not covered by the 10 vegetation types, reaching more than 0.9 at most of the grid points (Figure R8). Moreover, only 11.6% of the global land area is not covered by 10 vegetation types, the percentage of uncovered areas is small and therefore introduces little error. This type of area is mainly found in North Africa, the Middle East and parts of Central Asia. In summary, in the areas not covered by the 10 vegetation types, it is reasonable for us to use an equal weighting approach to fuse all ET datasets.

[Figure]

Figure R8 (Corresponding to Figure S36). Spatial distribution of multi-year average ET during 1980–2020 in the uncovered areas of 10 vegetation types. (a) the spatial distribution of ET based on the equal weight method, (b) the spatial distribution of ET based on the BTCH, (c) the correlation coefficient of ET between two methods, with the black dots representing that the ET of the grid point passed the significance test ($p<0.05$).

We have made the following additions in section 4.1 and Figure S36.

11. In Figure 4, 30 ET datasets were well evaluated, and Table 3 showed the guidelines for the use of ET datasets. So, in the BMA-ET fusion process, were all 30 ET datasets used for fusion, or only the recommended datasets used for fusion? If as

the authors stated, the accuracies of RA and LSM are not good, why are they still used for fusion?

**Response:** Thank you for your comment. The BMA method will control the weights of different ET datasets. Bayesian model averaging (BMA) provides a methodology to explicitly handle conceptual model uncertainty in the interpretation and analysis of environmental systems. This method combines the predictive capabilities of multiple different models and jointly assesses their uncertainty. The probability density function (pdf) of the quantity of interest predicted by BMA is essentially a weighted average of individual pdf's predicted by a set of different models that are centered around their forecasts. The weights assigned to each of the models reflect their contribution to the forecast skill over the training period.

Based on the results of ET integrated evaluation, the performance of the datasets based on remote sensing inversion and machine learning reconstruction is relatively better. Therefore, we also selected eight ET datasets for the fusion study as a robustness test of the number of ET datasets involved in the fusion process. The eight ET datasets include four datasets based on remote sensing inversion (PML, GLEAM, GLASS and PLSH datasets) and four datasets based on machine learning reconstruction (FLUXCOM-CRUNCEP_v8, FLUXCOM-GSWP3, FLUXCOM-WFDEI and MTE datasets). Only the better 8 ET data were selected for fusion, and there was little difference between the results of the two when all 30 ET data were involved in the fusion (Figure R9). The correlation coefficient between the 2 sets of ET fusion products exceeds 0.9 in most regions of the globe. This indicates that the BMA method gives higher weights to the better ET products. This also fully reflects the advantages of the BMA method. In order to prevent the effect of such a subjective behavior as selecting 8 out of 30 ET datasets, so this study still maintains the participation of all ET datasets.

[Figure]

Figure R9. Spatial distribution of multi-year average ET during 1981–2018 over the globe. (a) the spatial distribution of ET fusion product from 8 sets of ET data, (b) the spatial distribution of ET fusion product from 30 sets of ET data, (c) the correlation coefficient of two ET fusion products, with the black dots representing that the ET of the grid point passed the significance test ($p<0.05$).

12. In lines 237-238, the RS and ML ET datasets are recommended in the site scale validation results. Whereas, in lines 256-257, the ML ET datasets have greater TCH relative uncertainty. Do these two conclusions conflict? Please provide a detailed explanation.

**Response:** Thank you for your comment. The results show that the ML ET datasets have greater TCH relative uncertainty. This conclusion is for the basin scale. The basin-scale TCH analyses were performed based on the results of the basin-weighted average, rather than on a grid- by-grid basis within the basin. The results of basin

averaging make it difficult to accurately characterize the true situation at the global grid scale. Therefore, we also used the TCH method to analyze the uncertainty over the globe on a grid-by-grid basis. For TCH relative uncertainty, machine learning– and remote sensing–based datasets have less relative uncertainty (Figure R10). The results of the global TCH analysis are consistent with the results of the site-based ET assessment, both of which concluded that the remote sensing and machine learning-based ET dataset has higher accuracy. Therefore, there is no conflict between the conclusions.

[Figure]

Figure R10 (Corresponding to Figure S24). Spatial distribution of relative uncertainty of TCH for each ET dataset on monthly scale for the common coverage years from 1982 to 2011 (unit: %).

13. In Figure 1, the common period of coverage for all ET datasets is 1982–2011. How did this study produce the BMA-ET dataset from 1980 to 2020?

**Response:** This issue is similar to the fifth comment you mentioned, so see the response to the fifth comment for more information. Since different ET datasets cover different years, for the common coverage years 1982–2011, the weights of each ET dataset under each vegetation type were obtained by performing BMA analyses based on 30 ET datasets, and these weights were applied to all the years 1982–2011. For the non-common coverage years 1980–1981 and 2012–2020, the weights for each year were obtained by filtering all ET datasets covering that year for BMA analysis to obtain the corresponding weights.

We have made the following additions in section Text S6 and Figure S5.

14. In lines 355-356, the study recommended RS and ML based ET datasets (especially MTE and PML) based on the evaluation results. So why does the BMA-ET merge 30 ET datasets? Is it better to merge only MTE and PML?

**Response:** Thank you for your comment. This issue is similar to the eleventh comment you mentioned, so see the response to the eleventh comment for more information. The BMA method will control the weights of different ET datasets. Only the better 8 ET data were selected for fusion, and there was little difference between the results of the two when all 30 ET data were involved in the fusion. In order to prevent the effect of such a subjective behavior as selecting 8 out of 30 ET datasets, so this study still maintains the participation of all ET datasets. In addition, if only MTE and PML were fused, it would be difficult to fuse the two ET datasets to obtain ET data for a long time series (1980–2020) because the time periods covered by these two ET datasets are too short.

---------------------------------------------- end line----------------------------------------------

In order to make the review of our revision more convenient, we have marked all changes using the "Track Changes" function in Microsoft Word and have uploaded the "tracked changes" version as Supplementary Material.

**A DETAILED LIST OF RESPONSES**
**TO REVIEWER #2**

**Anonymous Reviewer #2 comments**

**General comments:**

The work "Multi-spatial scale assessment and multi-dataset fusion of global terrestrial evapotranspiration (ET) datasets" presents a detailed comparison of 30 global-scale evapotranspiration datasets and uses Bayesian Model Averaging to create a new weighted ensemble dataset. The paper is logically structured and clear overall. The comparison of such a large sample of ET datasets and the evaluation and comparison at a range of scales alone are interesting, novel, and valuable. For the dataset to be useful, more methodological details are needed about the pre-processing methods used align all datasets to a consistent spatial and temporal basis beyond the descriptions provided in the supplementary material. In addition, more detail is needed regarding the robustness of the BMA approach to key assumptions, namely land cover change, land cover classification uncertainty at the resolutions presented, and BMA model validation. Many such questions would be far easier to review and provide feedback on if (annotated) code used to generate the dataset were provided. These additions as well as a correction of the 1 degree resolution dataset are recommended before publication in ESSD.

**Response:** We greatly appreciate your careful reading of the manuscript, insightful comments, and valuable suggestions. Your thoughtful review has enhanced our paper considerably. The manuscript has been revised thoroughly according to your comments and those of the individual reviewers, with our point-by-point responses detailed below.

**Specific comments:**

1. There is a problem with the 1 degree dataset starting at approximately timestep 262.

**Response:** First of all, we thank you for your comments. We apologize that due to our carelessness, an incorrect version of the 1 degree spatial resolution BMA-ET dataset was uploaded here, for which we have updated and corrected.

2. Figure 7: What resolution and spatial interpolation methods were used to fill in data gaps in producing Figure 7? Below is a detail mean annual derived from 0.5 degree data (see attached notebook for full size image, the 500px restriction on image attachments is rather limiting here).

**Response:** Thank you for your comments. We apologize that we did not explain this clearly in the paper. In performing the plotting Figure 7, we assigned the grid points with no values to 0 mm and did not use any spatial interpolation to fill in the data gaps. We have corrected Figure 7.

3. Line 178: how was training/validation split done for evaluating BMA performance? Given the sparsity of flux sites, why wasn't cross-validation considered? How sensitive are model weights to the training sample?

**Response:** Many thanks for highlighting this point. In this study, we use 60% of station data for training and 40% of station data for validation. In order to verify the robustness of the results, we performed 2 additional sets of experiments, 70% of data for training and 30% of data for validation, 80% of data for training and 20% of data for validation, respectively. The results show that the data accuracy of the fusion product BMA-ET is not sensitive to FLUXNET station split. The accuracy evaluation results showed remarkable consistency across different training set proportions (60%, 70%, and 80%), with BMA-ET demonstrating correlation coefficients of 0.68, 0.67, and 0.65 respectively when compared to FLUXNET-ET (Figure R11). In addition, we also evaluate the accuracy of BMA-ET under different vegetation types. The results show that the accuracy of BMA-ET under each vegetation type is not sensitive to the split ratio of the training set (Figures R12–R14).

[Figure]

Figure R11 (Corresponding to Figure S29). Accuracy evaluation of BMA-ET. (a) 60% of data for training and 40% of data for validation, (b) 70% of data for training and 30% of data for validation, (c) 80% of data for training and 20% of data for validation.

[Figure]

Figure R12 (Corresponding to Figure S30). Accuracy evaluation of BMA-ET over different vegetation types. 60% of data for training and 40% of data for validation.

[Figure]

Figure R13 (Corresponding to Figure S31). Accuracy evaluation of BMA-ET over different vegetation types. 70% of data for training and 30% of data for validation.

[Figure]

Figure R14 (Corresponding to Figure S32). Accuracy evaluation of BMA-ET over different vegetation types. 80% of data for training and 20% of data for validation.

We have made the following additions in section 4.1.

4. Line 191: Can you quantify or estimate distributions of typical land cover changes at the appropriate dataset resolution as a basic test of model sensitivity to the stationary land cover assumption?

**Response:** We sincerely thank the reviewer for noting this. We performed a dynamic comparison of the dataset MOD12Q1 for the three periods of 2001, 2010 and 2020. In addition, we also analyzed the percentage of global land area covered by 12 vegetation types. Refer to Question 9 from Reviewer #1 for our specific responses.

5. Line 245: It isn't clear why correlations here are based on mean annual values and elsewhere (Figure S18) based on monthly data, making it more difficult to interpret the different comparisons presented (i.e. site, basin, global scales).

**Response:** Thank you for your comments. We apologize for not explaining this clearly in the paper. In Figure S18, each small circle represents a basin. This figure is not a comparison of ET datasets at any time scale, and it is a comparison of ET for all basins. Therefore, each small circle represents a multi-year average ET value for that basin. In this figure, even when replaced with monthly average ET values for each watershed, the comparison of multi-year average evapotranspiration from 30 ET datasets and multi-year average observed evapotranspiration from basin water balance are unchanged.

We have made the following additions in the title of Figure S20.

6. Line 284 -- comparing typical MAE values with the stated trend, what is the uncertainty in the 0.21mm/yr trend line? How significant is the magnitude (and precision) of this trend compared to typical variability due to error?

**Response:** Thank you for your comments. Based on the fused dataset BMA-ET, we calculated the change trend of ET from 1980 to 2020. The global average ET change trend and its uncertainty is 0.65 (0.51–0.78) mm/yr (Figure R15). By calculating the signal-to-noise ratio (SNR) of the year-by-year ET data, it is found that the SNR is less than 1, i.e., the ET trend is smaller than the interannual variability, which indicates that the ET trend signal is weak or noisy. The SNR only reflects the ratio of the trend to the interannual noise, and cannot distinguish between a weak true climatic signal (the trend is small) and a large noise in the data (e.g., fluctuations introduced by the observation error or the interpolation of the missing values). The Mann-Kendall test is a nonparametric test that does not require data distribution assumptions (Kendall, 1948; Mann, 1945); it is effective in detecting linear or nonlinear monotonic trends (continuously rising/declining) and is suitable for long-term climate change analyses; and it is insensitive to extreme values and robust. Therefore, we used the

Mann-Kendall test to verify the significance of the trend. Despite the low SNR of the ET series, the Mann-Kendall test showed that the trend was significant ($p<0.01$), suggesting a persistent upward trend in the ET series.

[Figure]

Figure R15 (Corresponding to Figure S25). Interannual variations of BMA-ET during 1980–2020. The global land average results are calculated based on a weighted average of the global land area.

7.A basic attempt to replicate Figure S23 was unsuccessful. There is likely a simple explanation for the substantial offset (~20mm) but it is much more laborious to investigate without the full replication code. A copy of the code used to generate the figures presented in this review is attached.

**Response:** I'm sorry we didn't make that clear. We used the area-weighted average method in calculating the global average ET. Data are available in a regular longitude-latitude grid. Therefore, grid cells do not have an equal size, with smaller grid cells at higher latitudes. Hence, for calculating the land area fractions, we must assign a weight to each grid cell on the basis of size. Here we compute these weights ($w_i$) as the size of each grid cell at latitude $lat_i$ relative to the size of the largest grid cells located at the Equator, given by:

$$W_i = \left| \frac{\sin(lat_i + (0.5 \times res)) - \sin(lat_i - (0.5 \times res))}{\sin(0.5 \times res) - \sin(-0.5 \times res)} \right| \tag{1}$$

where $lat$ is the vector indicating the latitude of each grid cell center, and ranges from $-90 + (0.5 \times res)$ to $90 - (0.5 \times res)$ with increasing step or resolution denoted by $res$.

We have made the following additions in Figure S25.

8.Figure 8: What are the units these models are compared on? i.e. is standard deviation mm/year?   Was some kind of normalization/standardization done to make the reference dataset standard deviation exactly 1?

**Response:** Thank you. The Taylor diagram was first proposed by Taylor and is mainly used to evaluate the ability of different models to simulate a variable. The plot combines three evaluation metrics: the correlation coefficient, the root mean square error, and the ratio of the standard deviations of the simulated and observed fields on a single polar plot. See Taylor for specific formulas. Standardized Taylor diagram normalize the standard deviation and root mean square error to eliminate their physical units of measure. When the correlation coefficient is larger, the root-mean-square error is smaller, and the ratio of the standard deviation of the simulated values to that of the observed values tends to be closer to 1, it indicates that the simulation results are in good agreement with the measured data, i.e., the model simulation results are highly reliable. In this study, a standardized Taylor diagram was used for the comprehensive assessment of the global ET dataset.

We have made the following additions in section 2.2, Figure 8 and Text S5.

9.Figure 8: What is the advantage of the BMA-ET dataset over the GLDAS-VIC dataset, or other datasets with similar correlation, lower RMSE, and standard deviation closer to the reference dataset?

**Response:** Thank you for your comments. In Figure R16 (Corresponding to Figure 8), BMA-ET has the highest correlation coefficient with FLUXNET2015. While the RMSE of BMA-ET vs. FLUXNET 2015 site ET was not the lowest, the ET dataset

with the lower RMSE had a much lower correlation coefficient than BMA-ET, less than 0.6. In addition, the BMA-ET dataset has a longer time period of coverage than other datasets with similar correlations, lower RMSEs and standard deviations closer to the reference dataset. Therefore, BMA-ET is more suitable for long-term climate change studies.

[Figure]

Figure R16 (Corresponding to Figure 8). Standardized Taylor diagram of ET datasets at all stations from 1991 to 2011. The observation data are ET from FLUXNET2015.

10.Line 313: What is the sensitivity of model performance to typical differences / uncertainties introduced by spatial scale mismatch?

**Response:** Since the original spatial resolutions of the 30 datasets are different, and this study uses a bilinear interpolation method to unify them to the same resolution, this resampling process introduces some uncertainty. Therefore, this study compared the differences in ET data at 1° and 0.5° globally and under each vegetation type, respectively. The results show that correlation coefficients for ET datasets at 1° and 0.5° spatial resolution globally are more than 0.9 among all ET datasets (Figure R17). For each vegetation type, correlation coefficients for ET datasets at 1° and 0.5° spatial resolution are over 0.8. This suggests that the uncertainty due to differences in spatial resolution is relatively small.

[Figure]

Figure R17 (Corresponding to Figure S4). Correlation coefficients for ET datasets at 1° and 0.5° spatial resolution globally and for each vegetation type.

We have made the following additions in section 2.2 and Figure S4.

11.Section 4.2: More discussion of how data leakage was avoided is needed. How is training data independent of validation data in each comparison?

**Response:** We thank the reviewer for your valuable suggestion. To address the above issues, we completed the following two components: 30 datasets are clustered and then fused; independent data source validation. Refer to Question 1 from Reviewer #1 for our specific responses.

---------------------------------------------- end line-------------------------------------------------

In order to make the review of our revision more convenient, we have marked all changes using the "Track Changes" function in Microsoft Word and have uploaded the "tracked changes" version as Supplementary Material.

**A DETAILED LIST OF RESPONSES**
**TO REVIEWER #3**

**Anonymous Reviewer #3 comments**

**General comments:**

This study applies BMA to merge multiple ET datasets, but two fundamental issues must be addressed before publication in ESSD: potential data leakage and the assumption of independent errors among datasets. If unresolved, the study lacks the necessary rigor for acceptance.

**Response:** We greatly appreciate your careful reading of the manuscript, insightful comments, and valuable suggestions. Your thoughtful review has enhanced our paper considerably. The manuscript has been revised thoroughly according to your comments and those of the individual reviewers, with our point-by-point responses detailed below.

**Specific comments:**

1. ML datasets (e.g., FLUXCOM, MTE) trained on FLUXNET are also used for BMA likelihood estimation, raising concerns about inflated weights. Has the author evaluated this effect? Clearly identify datasets incorporating FLUXNET and assess their influence on BMA weights. If necessary, limit their posterior weights or introduce independent validation datasets.

**Response:** First of all, we thank you for your comments. We used independent data sources to validate the fusion dataset BMA-ET, specifically including AmeriFlux, ChinaFlux, and ICOS. Refer to Question 1 from Reviewer #1 for our specific responses.

2.BMA assumes independent errors, but ML datasets share training data, RS datasets rely on MODIS, and LSMs use similar climate forcings. Has the author assessed inter-dataset correlations and their impact on weight allocation?

**Response:** We appreciate the reviewer's insightful comment. Indeed, BMA's assumption of independent errors may not fully hold for our 30 ET datasets, as they share underlying inputs (e.g., MODIS reflectance for RS products) or climate forcings (e.g., ERA5 for LSMs). We addressed this issue through the following approaches: 30 datasets are clustered and then fused. Refer to Question 1 from Reviewer #1 for our specific responses.

3.Considering introduce a covariance matrix (compute Pearson correlation matrices for FLUXNET residuals) into the likelihood function of BMA accounts for inter-dataset correlations. Compare weight distributions before and after adjustment. Alternatively, cluster highly correlated datasets (e.g., FLUXCOM, MTE) and down weight them collectively.

**Response:** We fully agree that accounting for inter-dataset correlations is critical for robust Bayesian Model Averaging (BMA). To address this, we completed the following: 30 datasets are clustered and then fused. Refer to Question 1 from Reviewer #1 for our specific responses.

4.The study applies a 60%-40% FLUXNET station split for BMA training and validation. Considering implement K-fold cross-validation or leave-one-out validation to assess the stability of BMA weights across different training subsets.

**Response:** Thank you for your comments. In this study, we use 60% of station data for training and 40% of station data for validation. In order to verify the robustness of the results, we did 2 additional sets of experiments: 70% of data for training and 30% of data for validation, 80% of data for training and 20% of data for validation, respectively. Refer to Question 3 from Reviewer #2 for our specific responses.

5.Bootstrap resampling of FLUXNET data could estimate confidence intervals for BMA weights and ET estimates. If dataset dependencies are strong, current uncertainty estimates may be underestimated. Try add confidence intervals (e.g., 95% CI) to BMA-ET results in figures such as Fig. 7 or Fig. S23 and discuss implications.

**Response:** We sincerely appreciate the reviewer's insightful suggestion regarding uncertainty quantification. Following the reviewer's advice, we performed bootstrap resampling based on the fused dataset BMA-ET, and we calculated the trend of ET change from 1980 to 2020. The global average ET change trend and its confidence intervals are 0.65 (0.51–0.78) mm/yr. We added shaded 95% CI bands to BMA-ET trend lines in Figure S25.

Implications:

The ET trend (0.65 mm/yr) indicates a significant increase in global land-averaged evapotranspiration (ET), which may be related to climate warming, longer vegetation growing seasons, or changes in water availability. The width of the 95% confidence intervals for the global terrestrial mean ET trend reflects data heterogeneity and methodological uncertainty (e.g., uneven distribution of FLUXNET sites). The main sources of uncertainty are input data and natural variability. The uncertainty of input data mainly involves systematic bias of different ET products in different climate zones (Jung et al., 2019). The natural variability refers inter-annual fluctuations, such as ENSO may mask long-term trends (Börgel et al., 2020; Grothe et al., 2020; Wu et al., 2011).

6.SI L355, TableS1, The TRENDY model dataset link is inaccessible. Is it publicly available? Clarify access restrictions and provide an alternative link if possible.

**Response:** We apologize that the previous link is no longer working, the TRENDY model data access is supplemented in Table S1.

7.L125, Table2, the citation Tian et al. (2015) may not be the best reference for DLEM ET data. Consider citing Pan et al. (2015) or Friedlingstein et al. (2023) (TRENDY v12 reference), which are more relevant to ET estimates.

**Response:** We have updated the references for the DLEM ET data.

8.L156, Suggest use FLUXNET2015 (2012-2014) data to supplement site validation and evaluate BMA-ET performance. Additionally, explore AmeriFlux or ICOS (2015-2020) data for further validation, enhancing extended-period credibility.

**Response:** Thanks. Following your suggestion, we used independent data sources to validate the fusion dataset BMA-ET, specifically including AmeriFlux, ChinaFlux, and ICOS. Additionally, we also evaluated the accuracy of BMA-ET using FLUXNET2015 data from 2012 to 2015. Refer to Question 1 from Reviewer #1 for our specific responses.

9.L274, Figure 6(a) color scheme appears cluttered. Align with subplot (b) by using consistent colors—RS datasets in red, ML datasets in yellow, etc. This improves clarity and direct comparison.

**Response:** Good advice. We have modified the colors in Figure 6 accordingly.

10.L300, figure8, same as above.

**Response:** We have modified the colors in Figure 8 accordingly.

-------------------------------------------- end line--------------------------------------------

In order to make the review of our revision more convenient, we have marked all changes using the "Track Changes" function in Microsoft Word and have uploaded the "tracked changes" version as Supplementary Material.

**References**

Börgel, F., Frauen, C., Neumann, T., and Meier, H. E. M.: The Atlantic Multidecadal Oscillation controls the impact of the North Atlantic Oscillation on North European climate, Environ. Res. Lett., 15, 104025, https://doi.org/10.1088/1748-9326/aba925, 2020.

Cai, S., Liu, D., Sulla-Menashe, D., and Friedl, M. A.: Enhancing MODIS land cover product with a spatial–temporal modeling algorithm, Remote Sensing of Environment, 147, 243–255, https://doi.org/10.1016/j.rse.2014.03.012, 2014.

Castelvecchi, D.: Can we open the black box of AI?, Nature News, 538, 20, https://doi.org/10.1038/538020a, 2016.

Grothe, P. R., Cobb, K. M., Liguori, G., Di Lorenzo, E., Capotondi, A., Lu, Y., Cheng, H., Edwards, R. L., Southon, J. R., Santos, G. M., Deocampo, D. M., Lynch-Stieglitz, J., Chen, T., Sayani, H. R., Thompson, D. M., Conroy, J. L., Moore, A. L., Townsend, K., Hagos, M., O'Connor, G., and Toth, L. T.: Enhanced El Niño–Southern Oscillation Variability in Recent Decades, Geophysical Research Letters, 47, e2019GL083906, https://doi.org/10.1029/2019GL083906, 2020.

Jung, M., Koirala, S., Weber, U., Ichii, K., Gans, F., Camps-Valls, G., Papale, D., Schwalm, C., Tramontana, G., and Reichstein, M.: The FLUXCOM ensemble of global land-atmosphere energy fluxes, Sci Data, 6, 74, https://doi.org/10.1038/s41597-019-0076-8, 2019.

Kendall, M. G.: Rank correlation methods, Griffin, Oxford, England, 1948.

Mann, H. B.: Nonparametric Tests Against Trend, Econometrica, 13, 245–259, https://doi.org/10.2307/1907187, 1945.

Vrugt, J. A. and Robinson, B. A.: Treatment of uncertainty using ensemble methods: Comparison of sequential data assimilation and Bayesian model averaging, Water Resources Research, 43, https://doi.org/10.1029/2005WR004838, 2007.

Vrugt, J. A., Diks, C. G. H., and Clark, M. P.: Ensemble Bayesian model averaging using Markov Chain Monte Carlo sampling, Environ Fluid Mech, 8, 579–595, https://doi.org/10.1007/s10652-008-9106-3, 2008.

Wu, S., Liu, Z., Zhang, R., and Delworth, T. L.: On the observed relationship between the Pacific Decadal Oscillation and the Atlantic Multi-decadal Oscillation, J Oceanogr, 67, 27–35, https://doi.org/10.1007/s10872-011-0003-x, 2011.

Zhang, H., Zhang, L., and Jiang, Y.: Overfitting and Underfitting Analysis for Deep Learning Based End-to-end Communication Systems, in: 2019 11th International Conference on Wireless Communications and Signal Processing (WCSP), 2019 11th International Conference on Wireless Communications and Signal Processing (WCSP), 1–6, https://doi.org/10.1109/WCSP.2019.8927876, 2019.

Zhang, Q. and Zhu, S.: Visual interpretability for deep learning: a survey, Frontiers Inf Technol Electronic Eng, 19, 27–39, https://doi.org/10.1631/FITEE.1700808, 2018.